# Real-World Use and Effectiveness of Carfilzomib Plus Dexamethasone in Relapsed/Refractory Multiple Myeloma in Europe

**DOI:** 10.3390/cancers14215311

**Published:** 2022-10-28

**Authors:** Evangelos Terpos, Renato Zambello, Xavier Leleu, Thomas Kuehr, Sorina N. Badelita, Eirini Katodritou, Alessandra Brescianini, Tony Liang, Sally Wetten, Jo Caers

**Affiliations:** 1Department of Clinical Therapeutics, School of Medicine, National and Kapodistrian University of Athens, 11528 Athens, Greece; 2Hematology and Clinical Immunology Branch, Department of Medicine, University of Padua, 35128 Padua, Italy; 3Department of Hematology, University Hospital Center La Miletrie and Inserm, CIC 1402, 86000 Poitiers, France; 4Department of Internal Medicine IV, Academic Teaching Hospital Wels-Grieskirchen, 4600 Wels, Austria; 5Fundeni Clinical Institute, 72437 Bucharest, Romania; 6Department of Hematology, Theagenio Cancer Hospital, 54007 Thessaloniki, Greece; 7Amgen (Europe) GmbH, 6343 Rotkreuz, Switzerland; 8Parexel International, Taipei 11071, Taiwan; 9Amgen Ltd., Uxbridge UB8 1DH, UK; 10Department of Hematology, Liege University Hospital Center, 4000 Liege, Belgium

**Keywords:** real world, multiple myeloma, proteasome inhibitor, carfilzomib, relapsed/refractory

## Abstract

**Simple Summary:**

Multiple treatment regimens are approved for relapsed/refractory multiple myeloma (RRMM), but evidence of their real-world use is limited. This study included 701 patients across 11 countries in Europe and Israel between March 2017 and March 2020, and examined the real-world use of carfilzomib in adults with RRMM who received at least one prior line of therapy. Here, we describe the results for 271 patients who received carfilzomib and dexamethasone (Kd). The overall response rates (ORR) with Kd treatment were high, both overall (68.8%) and by treatment line. In patients who are refractory to lenalidomide, the ORR of 59.9% is an encouraging outcome. In patients refractory to anti-CD38 treatment, the ORR was higher for those receiving Kd in earlier (66.7%) than later lines (fourth line or later, 49.1%). Our study demonstrates that Kd is used effectively, with an acceptable safety profile in routine clinical practice.

**Abstract:**

This prospective, observational study examined the real-world use of carfilzomib across 11 European countries in adults with relapsed/refractory multiple myeloma (RRMM) who received at least one prior line of therapy. Carfilzomib and dexamethasone (Kd) use, effectiveness and safety were analyzed. In total, 271 patients received Kd among 701 adults enrolled. The median relative dose intensity of carfilzomib was 82.7% (20/56 mg/m^2^, twice weekly). The overall response rate (ORR) to Kd was 68.8% (95% confidence interval [CI], 62.7–74.5): 79.2% in second line (2L), 71.6% in third line (3L) and 63.1% in fourth line or later (4L+). The ORR was 59.9% (95% CI, 51.1–68.1) in the lenalidomide-refractory subgroup and 67.7% (95% CI, 48.6–83.3) in the not lenalidomide-refractory subgroup. In the anti-CD38 refractory subgroup, the ORR was 51.6% (95% CI, 38.6–64.5); ORRs were higher when Kd was received at 2L/3L (66.7%) than at 4L+ (49.1%). Overall, patients were treated for a median time of 7.7 months. One-fifth of patients reported treatment-related treatment-emergent adverse events (≥grade 3), with a safety profile consistent with previous clinical trials. This study demonstrated the real-world use, effectiveness and safety of Kd in patients with RRMM. Despite the increasing number of new therapeutic strategies to treat RRMM, Kd remains a safe and effective option, even for older, frail and lenalidomide- or anti-CD38 mAb-refractory patients.

## 1. Introduction

Multiple myeloma (MM) is the second most common hematologic cancer, with an estimated incidence in Europe of approximately five per 100,000 per year [1].

The introduction of multi-drug regimens including proteasome inhibitors (PIs), immunomodulatory imide drugs (IMiDs) and monoclonal antibodies (mAbs) has significantly improved outcomes for patients with MM by increasing the depth and duration of response [1,2,3,4]. Managing patients with relapsed/refractory MM (RRMM) remains challenging [1], because nearly all patients eventually relapse and an increasing number display disease that is refractory to multiple classes of agents [5]. Moreover, survival rates reduce as patients become refractory to an increasing number of different drug classes [2]. Interestingly, data on the real-world use and effectiveness of multi-drug regimens in different treatment lines are limited [5].

Carfilzomib, a second-generation PI [6], was approved by the European Medicines Agency in 2015 for use in combination with lenalidomide and dexamethasone (KRd) to treat adults with MM who have received at least one prior therapy [7,8]. Since approval, the indication has been expanded to include use of carfilzomib and dexamethasone (Kd) alone, based on interim results from the randomized, phase 3 ENDEAVOR clinical trial in patients with RRMM [7,9]. In 2021, the indication has been expanded to include use in combination with daratumumab and dexamethasone [7,10]. 

Real-world evidence is important for ascertaining the use, effectiveness and safety of a drug in clinical practice beyond the data generated in clinical trials [11]. Therefore, this study aimed to describe the real-world use of Kd and its effectiveness and safety in adults with RRMM. 

## 2. Materials & Methods 

### 2.1. Study Design

This real-world, prospective, observational, cohort study (ClinicalTrials.gov identifier NCT03091127) was conducted at 113 centers across 11 countries (Austria, Belgium, Bulgaria, Czech Republic, France, Greece, Israel, Italy, the Netherlands, Norway and Romania) between March 2017 and March 2020. 

Adults with MM who had received at least one carfilzomib dose in routine clinical practice were eligible. Patients were excluded if prescribed carfilzomib as part of a clinical trial or within a compassionate use program. Frailty score at baseline (carfilzomib initiation) was derived using an algorithm based on the sum of age score, modified Charlson Comorbidity Index score and Eastern Cooperative Oncology Group (ECOG) performance status score as previously described [12].

### 2.2. Data Collection

Data for this study were collected from the first dose of carfilzomib administration until 30 days after patients’ final dose or 18 months after treatment initiation, death, loss to follow-up, withdrawal of consent or end of study (31 March 2020), whichever occurred earlier. Baseline data and initial follow-up time on Kd treatment were collected retrospectively upon enrollment, with prospective data collected at quarterly intervals thereafter until the end of the study. Treatment-emergent adverse events (TEAEs) of grade three and above were collected.

### 2.3. Outcomes

Study measures included Kd utilization and treatment characteristics, patient demographics and disease characteristics, safety profile of Kd and response to Kd as assessed by the investigator. Exploratory outcomes included response to Kd per the International Myeloma Working Group (IMWG) criteria, as assessed by an investigator, and analysis of duration of response to carfilzomib exposure.

Use of Kd in clinical practice was also described by line of therapy and by the following subgroups: patients who were exposed/refractory to (i) lenalidomide and (ii) anti-CD38 antibodies, in any prior line.

### 2.4. Statistical Analysis 

Descriptive statistics were used to summarize the data. Two-sided 95% CIs are presented when appropriate, calculated using Wilson’s method.

Time to event endpoints (duration of response and time to treatment discontinuation) and follow-up time on treatment were estimated using Kaplan–Meier methodology. Staging of MM disease at carfilzomib initiation was calculated from collected laboratory values according to the International Staging System (ISS) [13]. Relative dose intensity (RDI) was calculated relative to the dosing regimen on the carfilzomib label. Patients were defined as having refractory disease if they met at least one of the following International Myeloma Workshop Consensus Panel 1 criteria: (i) best response to therapy was stable or progressive disease; (ii) reason for drug discontinuation was disease progression; (iii) date of relapse/progression was strictly after the start date and within 60 days after the stop date of the drug [14]. 

## 3. Results 

### 3.1. Patient and Disease Characteristics

In total, 701 patients were enrolled across 11 participating countries in Europe and Israel; of these, 271 patients (38.7%) received Kd and are included in this analysis. Baseline disease and patient characteristics at Kd initiation are summarized in Table 1. The median age was 68.0 years. Overall, 35.1% of patients (n = 95) had calculated ISS scores, of whom 65.3% (n = 62) had an ISS stage of II or III. More than half of patients (51.6%) were considered frail (Table 1).

### 3.2. Treatment History Overall and by Line of Therapy

Disease and patient characteristics were similar regardless of treatment line, with median age being the exception: this increased with later treatment lines. Overall, patients were pre-treated with a median of three prior treatment lines (Table 1). Only 21.8% of patients had a hematopoietic stem cell transplant (HSCT) prior to receiving Kd in second line (2L) of therapy. Nearly all patients (97.0%) had received a prior treatment regimen that included bortezomib, to which more than half (54.4%) were refractory at Kd initiation (Table 1). 

### 3.3. Treatment Response Overall and by Line of Therapy

Of the 271 patients receiving Kd, 250 (92.3%) had a disease response assessment. The overall response rate (ORR) was 68.8% (95% CI, 62.7–74.5; n = 172), based on investigator assessment. For 97.2% (243 out of 250) of patients, the investigators’ disease response assessment was based on the IMWG criteria. The best overall response among patients who had a disease response assessment was a complete response (CR) or better in 13.2% of patients (n = 33) and a very good partial response (VGPR) or better in 43.6% (n = 109) of patients (Table 2). The ORR according to therapy line was 79.2%, 71.6% and 63.1% for patients treated with Kd in 2L, third line (3L) and fourth line or later (4L+), respectively (Table 2).

Among patients who achieved an overall response (n = 172), the observed overall median duration of response was 15.0 months (95% CI, 11.9–not estimable [NE]). For patients treated with Kd in 4L+, the median duration of response was 15.0 months; however, it was not estimable for patients who had received Kd in 2L or 3L.

### 3.4. Carfilzomib Administration and Discontinuation

Carfilzomib dosing was variable across the Kd cohort. For 77.5% of patients (n = 210), the European label dose and schedule (20/56 mg/m^2^ twice weekly) was planned to be administered [7]. For 12.2% of patients (n = 33), a dose and schedule of 20/27 mg/m^2^ twice weekly was planned. A weekly dosing schedule was planned for only 4.1% of patients but was used for 14.4% of patients. In a small proportion of patients (1.5%), a schedule of every 2 weeks was reported at some point during the treatment period.

The median (range) average dose of carfilzomib per administration across all doses administered was 52.1 (12.0–70.0) mg/m^2^ (56.0 [12.0–70.0] mg/m^2^ when excluding doses on days 1 and 2 of cycle 1). In total, 54.6% of patients received an RDI of 80% or higher; the median RDI for the total cohort throughout the study period was 82.7% (range 14.0–113.9).

During the study period, 76.0% of patients (n = 206) discontinued carfilzomib (median follow-up of 17.5 months [95% CI, 16.8–17.9]) (Figure 1). The median (95% CI) time to discontinuation was 7.7 (6.5–9.0) months overall, and 9.7 (6.9–13.4), 8.6 (6.3–10.6) and 6.9 (5.4–8.3) months for patients receiving Kd at 2L, 3L and 4L+, respectively. Among patients who discontinued carfilzomib, the main reasons for discontinuation were disease progression/refractory disease (51.9%), adverse events (AEs) (20.4%) and required level of treatment response achieved (9.7%).

### 3.5. Lenalidomide-Exposed Subgroups

#### 3.5.1. Patient and Disease Characteristics in the Lenalidomide Subgroups

Among the 271 patients who received Kd, 185 (68.3%) had previously been treated with lenalidomide. Of these, 81.6% (n = 151) patients presented with lenalidomide-refractory disease and 18.4% (n = 34) with disease not refractory to lenalidomide (Table 3). In the refractory subgroup, 46 patients (30.5%) received Kd at 2L/3L and 105 (69.5%) received Kd at 4L+. In the not-refractory subgroup, 12 patients (35.3%) received Kd at 2L/3L and 22 (64.7%) received Kd at 4L+.

At Kd initiation, median age was 70.0 years in the refractory subgroup (2L/3L, 70.0 years; 4L+, 69.0 years) and 71.0 years in the not-refractory subgroup (2L/3L, 71.0 years; 4L+, 70.5 years) (Appendix A). 

Among patients for whom the frailty score could be derived, 46.8% (n = 44 out of 94 patients) in the refractory subgroup and 60.9% (n = 14 out of 23 patients) in the not-refractory subgroup were considered frail (ie, patients with a frailty score sum ≥2). In the refractory subgroup, 39.1% (n = 9 out of 23) of patients at 2L/3L were classed as frail; this increased to almost a half (49.3%, n = 35 out of 71 patients) at 4L+. For the patients in the not-refractory subgroup, a smaller proportion were considered frail among those initiating Kd at later therapy lines than at earlier lines (66.7%, n = 6 out of 9 patients at 2L/3L vs. 57.1%, n = 8 out of 14 patients at 4L+) (Appendix A).

#### 3.5.2. Treatment History of Lenalidomide Subgroups by Line of Therapy

At Kd initiation, patients in both subgroups had been exposed to a median of three prior treatment lines (Appendix A). More than two-thirds of patients in the lenalidomide-refractory subgroup were double-class refractory (43.0%) or triple-class refractory (29.8%) to previous PI, IMiD and anti-CD38 mAb therapies (Appendix A). Notably, 41.2% of patients with disease not refractory to lenalidomide presented with disease also not refractory to any previous therapies (Appendix A). In both subgroups, a similar proportion of patients underwent HSCT (refractory, 52.3% vs. not refractory, 52.9%) before initiating Kd. 

#### 3.5.3. Response by Lenalidomide Subgroups and by Line of Therapy

Among patients with a disease response assessment (refractory, n = 137 [90.7%] and not refractory, n = 31 [91.2%]), ORR was 59.9% (95% CI, 51.1–68.1) and 67.7% (95% CI, 48.6–83.3) in the refractory and not-refractory subgroups, respectively (Table 3). In the refractory subgroup, 9.5% had a CR or better and 32.1% had a VGPR or better compared with 16.1% and 41.9% in the not-refractory subgroup, respectively (Table 3).

For patients receiving Kd at 2L/3L and with a disease response assessment (n = 44 in the refractory and n = 11 in the not-refractory subgroups), ORR was 59.1% (95% CI, 43.2–73.7) and 72.7% (95% CI, 39.0–94.0) in the refractory and not-refractory subgroups, respectively. When receiving Kd at 4L+ (refractory, n = 93; not refractory, n = 20), ORR was 60.2% (95% CI, 49.5–70.2) and 65.0% (95% CI, 40.8–84.6) in the refractory and not-refractory subgroups, respectively (Table 3).

#### 3.5.4. Carfilzomib Use and Discontinuation in Lenalidomide Subgroups

The median time to carfilzomib discontinuation was similar between patients in the refractory subgroup and those in the not-refractory subgroup (6.3 months [95% CI: 4.8–7.7] and 7.0 months [95% CI: 5.0–14.3], respectively). The main reasons for carfilzomib discontinuation were aligned with the overall population analysis for both subgroups. 

When receiving Kd at 2L/3L, the median (95% CI) time to carfilzomib discontinuation was 6.4 (3.6–9.3) months in the refractory subgroup and 7.9 (0.5–NE) months in the not-refractory subgroup. When receiving Kd at 4L+, the median (95% CI) time to carfilzomib discontinuation was 6.3 (4.1–7.9) months in the refractory subgroup and 6.9 (5.0–14.3) months in the not-refractory subgroup.

### 3.6. Anti-CD38 mAb-Refractory Subgroup 

#### 3.6.1. Patient and Disease Characteristics

A total of 73 patients had previously received anti-CD38 mAb treatment: daratumumab was used in 70 patients and isatuximab in three patients (Table 1). Most patients (n = 71; 97.3%) had anti-CD38 mAb-refractory disease at carfilzomib initiation. The median age was 69.0 years (2L/3L, 70.0 years; 4L+, 68.0 years). More than half of patients with a derived frailty score were classified as frail (55.3%). The proportion of frail patients increased in later lines (2L/3L, 33.3% vs. 4L+, 58.5%) (Appendix A).

#### 3.6.2. Treatment History Overall and by Line of Therapy

Overall, within this subgroup, most patients initiated Kd at 4L+ (2L/3L, 12.7%; 4L+, 87.3%) (Appendix A). Three patients had received anti-CD38 treatment in more than one prior line. Among patients who were refractory to anti-CD38 mAb therapy and had received it as continuous therapy in any prior line, 52.1% received it as monotherapy. Most of these patients were triple-class refractory (69.0%) and 23.9% were double-class refractory. Overall, 56.3% of patients underwent a HSCT before initiating Kd (Appendix A).

#### 3.6.3. Response Overall and by Line of Therapy

The ORR (95% CI) was 51.6% (38.6–64.5) overall, 66.7% (29.9–92.5) at 2L/3L and 49.1% (35.1–63.2) at 4L+ (Table 4). A VGPR or better was recorded for 27.4% of patients in this subgroup; the proportion of patients achieving a VGPR or better was higher among those initiating Kd at earlier rather than later treatment lines (2L/3L, 44.4% vs. 4L+, 24.5%).

#### 3.6.4. Carfilzomib Discontinuation

The median time to carfilzomib discontinuation in this subgroup was 4.9 months (95% CI, 3.3–6.9). The main reasons for carfilzomib discontinuation were disease progression/refractory disease (57.9% of those who discontinued), AEs (29.8%) and death (7.0%). For patients receiving Kd at 2L/3L and 4L+, the median (95% CI) time to discontinuation was 4.9 (1.8–NE) and 4.8 (3.3–7.2) months, respectively.

### 3.7. Safety

#### 3.7.1. Safety in the Overall Kd Population and by Treatment Line

TEAEs (≥grade 3) were reported for 122 patients (45.0%) receiving Kd. A total of 56 patients (20.7%) reported treatment-related TEAEs (Table 5). Overall, the most common treatment-related TEAEs were thrombocytopenia (5.2%) and hypertension (4.1%) (Table 5). Treatment-related serious adverse events (SAEs) occurred in 38 patients (14.0%). For 16 patients (5.9%), treatment-related TEAEs led to discontinuation of carfilzomib.

Patients who initiated Kd in 4L+ experienced a greater proportion of treatment-related TEAEs (24.7%) than patients who initiated Kd in 2L (10.9%) or 3L (20.0%); they also experienced a greater proportion of treatment-related SAEs (17.1% vs. 7.3% and 12.9%, respectively) and events leading to carfilzomib discontinuation (10.3% vs. 1.8% and 0.0%, respectively). All cardiac events occurred in patients initiating Kd in 4L+ (Table 5). Two treatment-related fatal AEs (one of cardiac arrhythmia and one of coronary artery disorder) occurred in patients who initiated Kd in 4L+ and had disease refractory to lenalidomide (Table 5, Appendix A).

#### 3.7.2. Safety in the Lenalidomide-Exposed Subgroups

Among the 185 patients who were exposed to lenalidomide (two-thirds of them initiated Kd in 4L+), TEAEs were reported in 76 patients (50.3%) in the refractory subgroup and in 14 patients (41.2%) in the not-refractory subgroup (Appendix A). A greater proportion of treatment-related TEAEs occurred in the refractory subgroup (24.5%) than in the not-refractory subgroup (20.6%), but did not translate into more SAEs (15.9% and 17.6%, respectively). In the refractory subgroup, the most common treatment-related TEAEs were thrombocytopenia (6.6%), hypertension (5.3%), anemia (4.6%) and heart failures (4.0%) (Appendix A). Treatment-related TEAEs led to carfilzomib discontinuation in 7.9% of patients in the refractory subgroup and 5.9% of patients in the not-refractory subgroup. 

#### 3.7.3. Safety in the Anti-CD38 mAb-Refractory Subgroup

In this small subgroup of 71 patients, comprised mostly of patients who initiated Kd in 4L+ (87.3%), TEAEs were reported for 43 patients (60.6%) (Appendix A). Treatment-related TEAEs occurred in 23.9% of patients, and none of these events were reported in patients who initiated Kd in 2L/3L. Among the 62 patients who initiated Kd in 4L+, treatment-related SAEs occurred in 12 patients and treatment-related TEAEs led to carfilzomib discontinuation in 7 patients. The treatment-related TEAE profile was similar to that of the overall Kd population and the 4L+ subset. The above-mentioned fatal event of coronary artery disorder occurred in one patient in this subgroup, who was also in the lenalidomide-refractory subgroup. 

## 4. Discussion

This real-world study describes the use, safety and effectiveness of Kd in patients with RRMM, including patients with lenalidomide-refractory and anti-CD38 mAb-refractory disease. The ORRs with Kd treatment were high, both overall (68.8%) and by therapy line (2L, 79.2%; 3L, 71.6%; 4L+, 63.1%). In addition, the safety profile of Kd was consistent with the known safety profiles of carfilzomib and dexamethasone individually [7,17]. 

The results from our study were largely similar to what is observed in the clinical trial setting with some notable differences. The ORRs in our study were similar or slightly lower than those observed in the phase 3 randomized controlled trial ENDEAVOR (68.8% vs. 77.0%); the proportion of patients achieving a CR or better was 13.2% versus 13.0%, respectively, and those achieving a VGPR or better were 43.6% versus 54.0%, respectively [9]. Notably, compared with ENDEAVOR, patients in our study were slightly older [9], and a higher proportion had an ECOG performance status of 2 or higher (19.8% vs. 7.0%) [9], an ISS score of III (37.9% vs. 24.6%) [18], and a greater proportion were considered frail (51.6% vs. 37.7%) [19]. Whereas the ENDEAVOR study compared Kd with bortezomib and dexamethasone in patients with RRMM who had received one to three previous treatments, our study enrolled patients who had received up to 10 treatment lines [9,18,19]. As such, the two studies are not directly comparable. Overall, the benefit—risk profile of Kd in the real world was consistent with clinical trial results from ENDEAVOR [20,21]. Our data showed that Kd achieved deeper responses in earlier lines of therapy, emphasizing the importance of optimizing treatment sequencing. Furthermore, our study aligns with previous reports that patients in clinical practice are often not as healthy as a clinical trial population [22].

Consistent with this, two further recently published real-world studies found Kd to be safe and effective in patients with RRMM, albeit with lower ORRs than those observed in ENDEAVOR. In a retrospective analysis of 75 patients from Italy—two-thirds of whom were over the age of 65 at treatment initiation—Kd was associated with an ORR of 60% and median PFS of 10 months [23]. Notably, survival was unaffected by whether patients were refractory to their previous treatment or to lenalidomide [23]. In a real-world post-marketing surveillance study in Japan, the ORR in Kd-treated patients with RRMM (median age 71 years) was 52.9% [24]. This lower ORR compared to ENDEAVOR may reflect the fact that 33.5% of patients in the Japanese study had an ECOG PS ≥ 2, whereas in ENDEAVOR, patients with ECOG PS > 2 were excluded [9]. While these real-world findings are generally consistent with those of our study (in which 20% of patients were ECOG PS ≥ 2), they also serve to highlight the limited generalizability between randomized clinical trials and real-world data.

Owing to the paucity of published studies relating to patients with RRMM, our subgroup analyses of this heavily pre-treated patient population are of particular interest. The ORR in the lenalidomide-refractory subgroup was 59.9%, which is an encouraging outcome in this subset of patients. Patients with anti-CD38 mAb-refractory disease had an ORR of 51.6% overall, and the ORR was somewhat higher for those receiving Kd in earlier treatment lines (2L/3L, 66.7%; 4L+, 49.1%). Patients with anti-CD38 mAb-refractory disease often have a poor prognosis and few data are available regarding viable treatment options for them. Our ORR findings are complementary to the results from a multicenter, retrospective study in patients with anti-CD38 mAb-refractory disease, which reported an ORR of 32.3% among patients who received a carfilzomib-based regimen in the treatment line following an anti-CD38 mAb [2]. In addition, it was also found that both carfilzomib-based therapy and daratumumab plus IMiD regimens were associated with a reduced risk of progression or death, compared with other treatment groups analyzed in the study. Hence, our data provide additional support for the use of Kd as a treatment option in this patient group. 

Lenalidomide administration is standard of care as first-line therapy for patients who are ineligible for a transplant [25] and it is often used as maintenance for patients who are eligible for a transplant as first-line therapy [26]. Daratumumab use is likely to increase with the approval of its use in combination therapy in first line (daratumumab–lenalidomide–dexamethasone and daratumumab–bortezomib–melphalan–prednisone) and with the European Hematology Association and European Society for Medical Oncology guidelines recommending its use in this setting [27,28]. Therefore, patients who are refractory to these drugs have been identified as a potential future unmet need owing to the likelihood of future disease relapse and consequent need for further therapy [2]. Notably, our study showed that most patients who received lenalidomide or daratumumab in a previous treatment line had become refractory by the time they initiated Kd in 4L+. The benefit–risk profile of Kd in these refractory subgroups was similar to the overall profile in Kd patients who started carfilzomib in 4L+.

Our study provides important information regarding the real-world use of carfilzomib. Data showed that the planned dosing of carfilzomib varied from the EU prescribing information for a number of patients. Clinical sites planned to administer the label dose of 20/56 mg/m^2^ to 77.5% of patients and a dose of 20/27 mg/m^2^ to 12.2% of patients; however, only 54.6% of patients achieved an RDI of 80% or higher. Furthermore, despite only a small proportion of patients (4.1%) being scheduled for weekly treatment, a noticeably higher proportion (14.4%) had weekly dosing in practice. This observed dosing schedule reduction may have been for convenience or to reduce the treatment burden on the patient [29]. In our study, patients receiving Kd were treated for a median time of 7.7 months overall, which can be explained by the large proportion of patients who started Kd in 4L+ (53.9%). Carfilzomib discontinuation was mainly due to disease progression, which has been previously reported for patients treated with KRd [8,30].

Notable differences in the usage of Kd and KRd regimens were reported in a previous analysis from this real-world study, which showed that Kd is preferentially used in later lines than KRd (53.9% vs. 18.5% of patients in 4L+, respectively), to treat older and more heavily pre-treated patients with refractory disease [20]. In the KRd cohort of 383 patients, 60.1% of patients had received KRd in 2L [30]. Specific reimbursement policies limiting access to the KRd regimen in some Central Eastern European countries at the time of this study may explain the choice of Kd in patients in 2L and in patients with disease not refractory to lenalidomide [31]. 

Some limitations of this study exist. Our analysis is limited by a small sample size in some subgroups; by missing or incomplete data owing to the nature of information being collected from medical charts; by the absence of subgroup data according to high-risk disease features, such as high-risk cytogenetics; by the study’s observational and non-comparative design; and by the lack of central response evaluation. Nevertheless, data were captured from 113 oncology centers across 11 EU countries. As such, the conclusions could be considered more representative and generalizable to the wider real-world population of patients with RRMM than previously reported clinical trial data. 

## 5. Conclusions 

Our study demonstrates that Kd is used effectively and has an acceptable safety profile in routine clinical practice in patients with RRMM, with notable differences with the pivotal clinical trial population. The study provides clinicians with better insight regarding the use of Kd in patients with RRMM. Despite the increasing number of new therapeutic strategies for RRMM [1,27], Kd remains a safe and effective option, even for older, frail and lenalidomide- or anti-CD38 mAb-refractory patients, and those who have already been exposed to newer treatments. Future prospective studies with larger patient groups are warranted to better understand patient and disease predictors of Kd discontinuation in the relapsed/refractory setting. 

## Figures and Tables

**Figure 1 cancers-14-05311-f001:**
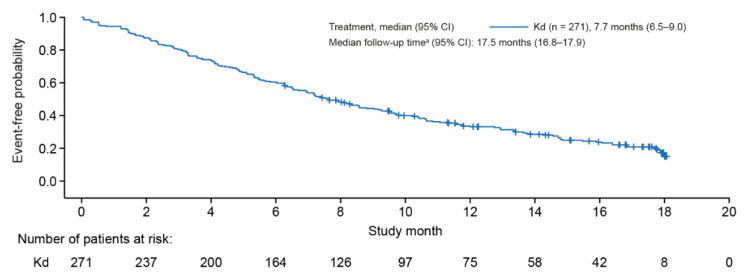
Kaplan–Meier estimates of overall time to discontinuation of carfilzomib for patients with MM receiving treatment with Kd. Time to discontinuation is defined as the time from carfilzomib initiation date to the date of treatment discontinuation. An event is defined as the discontinuation of carfilzomib. Patients who have not discontinued carfilzomib are censored on their last recorded non-zero dose date. ^a^ Estimated using reverse Kaplan–Meier method [16]. CI, confidence interval; Kd = carfilzomib combined with dexamethasone; MM = multiple myeloma.

**Table 1 cancers-14-05311-t001:** Baseline disease and patient characteristics and treatment history for study patients, overall and by line of therapy.

	2L (n = 55)	3L (n = 70)	4L+ (n = 146)	Overall (N = 271)
**Patients and disease characteristics**				
**Sex** Male	23 (41.8)	36 (51.4)	79 (54.1)	138 (50.9)
**Age at carfilzomib initiation** Mean, years (SD) Median (min, max), years <65 years 65–74 years ≥75 years	65 (8.2) 66 (47, 80) 25 (45.5) 22 (40.0) 8 (14.5)	69 (8.4) 70 (45, 84) 21 (30.0) 34 (48.6) 15 (21.4)	68 (8.1) 69 (50, 92) 48 (32.9) 68 (46.6) 30 (20.5)	68 (8.3) 68 (45, 92) 94 (34.7) 124 (45.8) 53 (19.6)
**ISS stage at carfilzomib initiation ^a^** I ^b^ II ^b^ III ^b^	20 (36.4) 9 (45.0) 5 (25.0) 6 (30.0)	23 (32.9) 7 (30.4) 6 (26.1) 10 (43.5)	52 (35.6) 17 (32.7) 15 (28.8) 20 (38.5)	95 (35.1) 33 (34.7) 26 (27.4) 36 (37.9)
**Patients with ECOG PS at carfilzomib** **initiation** 0–1 ^a^ 2–3 ^a^ 4 ^a^	44 (80.0) 35 (79.5) 9 (20.5) 0 (0.0)	46 (65.7) 38 (82.6) 8 (17.4) 0 (0.0)	102 (69.9) 81 (79.4) 21 (20.6) 0 (0.0)	192 (70.8) 154 (80.2) 38 (19.8) 0 (0.0)
**Patients with derived frailty score at** **carfilzomib initiation ^c^** Fit (0) ^a^ Intermediate (1) ^a^ Frail (≥2) ^a^	44 (80.0) 5 (11.4) 17 (38.6) 22 (50.0)	46 (65.7) 6 (13.0) 15 (32.6) 25 (54.3)	102 (69.9) 16 (15.7) 34 (33.3) 52 (51.0)	192 (70.8) 27 (14.1) 66 (34.4) 99 (51.6)
**Cytogenetic risk at diagnosis ^a^** High Standard Not available	55 (100) 4 (7.3) 2 (3.6) 49 (89.1)	70 (100) 6 (8.6) 5 (7.1) 59 (84.3)	146 (100) 22 (15.1) 10 (6.8) 114 (78.1)	271 (100) 32 (11.8) 17 (6.3) 222 (81.9)
**Treatment history**				
**Number of prior lines of treatment:** Median (min, max)	1 (1, 1)	2 (2, 2)	4 (3, 10)	3 (1, 10)
**Previous HSCT**	12 (21.8)	24 (34.3)	84 (57.5)	120 (44.3)
**Previously treated for MM in a clinical trial**	5 (9.1)	8 (11.4)	45 (30.8)	58 (21.4)
**Type of previous therapy**				
**Proteosome inhibitor ^d^**	51 (92.7)	70 (100.0)	145 (99.3)	266 (98.2)
Bortezomib	50 (90.9)	70 (100.0)	143 (97.9)	263 (97.0)
Carfilzomib	0 (0.0)	0 (0.0)	9 (6.2)	9 (3.3)
Ixazomib	1 (1.8)	4 (5.7)	14 (9.6)	19 (7.0)
**IMiD ^d^**	10 (18.2)	51 (72.9)	136 (93.2)	197 (72.7)
Lenalidomide	9 (16.4)	49 (70.0)	127 (87.0)	185 (68.3)
Pomalidomide	1 (1.8)	3 (4.3)	69 (47.3)	73 (26.9)
Thalidomide	1 (1.8)	13 (18.6)	59 (40.4)	73 (26.9)
**Monoclonal antibody ^d^**	1 (1.8)	9 (12.9)	66 (45.2)	76 (28.0)
Daratumumab	0 (0.0)	9 (12.9)	61 (41.8)	70 (25.8)
Elotuzumab	0 (0.0)	0 (0.0)	5 (3.4)	5 (1.8)
Isatuximab	1 (1.8)	0 (0.0)	2 (1.4)	3 (1.1)
**Refractory to any previous treatment line ^e,f^**				
Bortezomib	19 (38.0)	30 (42.9)	94 (65.7)	143 (54.4)
Daratumumab	0 (−)	8 (88.9)	60 (98.4)	68 (97.1)
Lenalidomide	7 (77.8)	39 (79.6)	105 (82.7)	151 (81.6)

Data presented as n (%) unless stated otherwise. ^a^ Percentage is relative to the number of patients with data. ^b^ Calculated from collected laboratory values. ^c^ Patients with frailty score sums of 0, 1 or ≥2 were classified as fit, intermediate or frail, respectively. ^d^ Patients may have received more than one drug within a given drug class. Hence, the total numbers reported for each drug class may be smaller than the sum of the individual values of each drug within that class. ^e^ A patient was classified as refractory to a drug by IMWG definition if they met at least one of the three following criteria: best response to any regimen containing the drug was either stable or progressive disease; the reason the treatment was stopped was progression in any regimen containing the drug; date of relapse/progression was after the start date and within 60 days (inclusive) after the stop date of the drug in any regimen containing the drug. ^f^ Percentage was calculated based on the number of patients who previously received the indicated treatment. 2L = second line; 3L, third line; 4L+ = fourth or later lines; ECOG PS = Eastern Cooperative Oncology Group performance status; HSCT = hematopoietic stem cell transplant; IMiD = immunomodulatory drug; IMWG = International Myeloma Working Group; ISS = International Staging System; Kd = carfilzomib combined with dexamethasone; MM = multiple myeloma; SD = standard deviation.

**Table 2 cancers-14-05311-t002:** Treatment response.

	2L (n = 55)	3L (n = 70)	4L+ (n = 146)	Overall (N = 271)
**Patients with disease response assessment**	53 (96.4)	67 (95.7)	130 (89.0)	250 (92.3)
**ORR ^a,b^** **[95% CI] ^c^**	42 (79.2) [65.9–89.2]	48 (71.6) [59.3–82.0]	82 (63.1) [54.2–71.4]	172 (68.8) [62.7–74.5]
**Best overall response ^b^** CR or better VGPR or better sCR CR VGPR PR MR SD PD NE	10 (18.9) 32 (60.4) 0 (0.0) 10 (18.9) 22 (41.5) 10 (18.9) 0 (0.0) 3 (5.7) 8 (15.1) 0 (0.0)	9 (13.4) 32 (47.8) 1 (1.5) 8 (11.9) 23 (34.3) 16 (23.9) 2 (3.0) 5 (7.5) 10 (14.9) 2 (3.0)	14 (10.8) 45 (34.6) 2 (1.5) 12 (9.2) 31 (23.8) 37 (28.5) 7 (5.4) 16 (12.3) 25 (19.2) 0 (0.0)	33 (13.2) 109 (43.6) 3 (1.2) 30 (12.0) 76 (30.4) 63 (25.2) 9 (3.6) 24 (9.6) 43 (17.2) 2 (0.8)

Data presented as n (%). ^a^ ORR is defined as the proportion of patients who have a best overall response of PR or better, ie, sCR, CR, VGPR or PR. ^b^ Percentage is relative to the number of patients with a disease response assessment. ^c^ 95% CIs were estimated using the Clopper–Pearson method [15]. 2L = second line; 3L = third line; 4L+ = fourth or later lines; CI = confidence interval; CR = complete response; Kd = carfilzomib combined with dexamethasone; MR = minimal response; NE = not evaluable; ORR = overall response rate; PD = progressive disease; PR = partial response; sCR = stringent CR; SD = stable disease; VGPR = very good PR.

**Table 3 cancers-14-05311-t003:** Treatment response to Kd for lenalidomide-exposed patients.

	Lenalidomide-Exposed: Refractory			Lenalidomide-Exposed: Not Refractory		
	2L/3L (n = 46)	4L+ (n = 105)	Overall (n = 151)	2L/3L (n = 12)	4L+ (n = 22)	Overall (n = 34)
**Patients with disease** **response assessment**	44 (95.7)	93 (88.6)	137 (90.7)	11 (91.7)	20 (90.9)	31 (91.2)
**ORR ^a,b^** **[95% CI] ^c^**	26 (59.1) [43.2–73.7]	56 (60.2) [49.5–70.2]	82 (59.9) [51.1–68.1]	8 (72.7) [39.0–94.0]	13 (65.0) [40.8–84.6]	21 (67.7) [48.6–83.3]
**Best overall response ^b^** CR or better VGPR or better sCR CR VGPR PR MR SD PD NE	5 (11.4) 16 (36.4) 0 (0.0) 5 (11.4) 11 (25.0) 10 (22.7) 1 (2.3) 5 (11.4) 11 (25.0) 1 (2.3)	8 (8.6) 28 (30.1) 1 (1.1) 7 (7.5) 20 (21.5) 28 (30.1) 5 (5.4) 13 (14.0) 19 (20.4) 0 (0.0)	13 (9.5) 44 (32.1) 1 (0.7) 12 (8.8) 31 (22.6) 38 (27.7) 6 (4.4) 18 (13.1) 30 (21.9) 1 (0.7)	2 (18.2) 6 (54.5) 0 (0.0) 2 (18.2) 4 (36.4) 2 (18.2) 0 (0.0) 0 (0.0) 2 (18.2) 1 (9.1)	3 (15.0) 7 (35.0) 1 (5.0) 2 (10.0) 4 (20.0) 6 (30.0) 1 (5.0) 3 (15.0) 3 (15.0) 0 (0.0)	5 (16.1) 13 (41.9) 1 (3.2) 4 (12.9) 8 (25.8) 8 (25.8) 1 (3.2) 3 (9.7) 5 (16.1) 1 (3.2)

Data presented as n (%). ^a^ The ORR is defined as the proportion of patients who have a best overall response of PR or better (sCR, CR, VGPR or PR). ^b^ Percentage is relative to the number of patients with a disease response assessment. ^c^ 95% CIs were estimated using the Clopper–Pearson method [15]. 2L = second line; 3L = third line; 4L+ = fourth or later lines; CI = confidence interval; CR = complete response; Kd = carfilzomib combined with dexamethasone; MR = minimal response; NE = not evaluable; ORR = overall response rate; PD = progressive disease; PR = partial response; sCR = stringent CR; SD = stable disease; VGPR = very good partial response.

**Table 4 cancers-14-05311-t004:** Treatment response for patients with anti-CD38 mAb-refractory disease.

	2L/3L (n = 9)	4L+ (n = 62)	Overall (n = 71)
**Patients with disease response assessment**	9 (100)	53 (85.5)	62 (87.3)
**ORR ^a,b^** **[95% CI] ^c^**	6 (66.7) [29.9–92.5]	26 (49.1) [35.1–63.2]	32 (51.6) [38.6–64.5]
**Best overall response ^b^** CR or better VGPR or better sCR CR VGPR PR MR SD PD NE	0 (0.0) 4 (44.4) 0 (0.0) 0 (0.0) 4 (44.4) 2 (22.2) 0 (0.0) 1 (11.1) 2 (22.2) 0 (0.0)	4 (7.5) 13 (24.5) 1 (1.9) 3 (5.7) 9 (17.0) 13 (24.5) 5 (9.4) 12 (22.6) 10 (18.9) 0 (0.0)	4 (6.5) 17 (27.4) 1 (1.6) 3 (4.8) 13 (21.0) 15 (24.2) 5 (8.1) 13 (21.0) 12 (19.4) 0 (0.0)

Data presented as n (%). ^a^ The ORR is defined as the proportion of patients who have a best overall response of PR or better (sCR, CR, VGPR or PR). ^b^ Percentage is relative to the number of patients with a disease response assessment. ^c^ 95% CIs were estimated using the Clopper–Pearson method [15]. 2L = second line; 3L = third line; 4L+ = fourth or later lines; CI = confidence interval; CR = complete response; Kd = carfilzomib combined with dexamethasone; mAb = monoclonal antibody; MR = minimal response; NE = not evaluable; ORR = overall response rate; PD = progressive disease; PR = partial response; sCR = stringent CR; SD = stable disease; VGPR = very good partial response.

**Table 5 cancers-14-05311-t005:** Summary of AEs.

	2L (n = 55)	3L (n = 70)	4L+ (n = 146)	Overall (N = 271)
**All CTCAE grade 3 and above TEAEs** SAEs AEs leading to discontinuation of carfilzomib Fatal AEs	21 (38.2) 17 (30.9) 5 (9.1) 5 (9.1)	27 (38.6) 20 (28.6) 2 (2.9) 4 (5.7)	74 (50.7) 62 (42.5) 26 (17.8) 14 (9.6)	122 (45.0) 99 (36.5) 33 (12.2) 23 (8.5)
**All CTCAE grade 3 and above treatment-related TEAEs** SAEs AEs leading to discontinuation of carfilzomib Fatal AEs ^a^	6 (10.9) 4 (7.3) 1 (1.8) 0 (0.0)	14 (20.0) 9 (12.9) 0 (0.0) 0 (0.0)	36 (24.7) 25 (17.1) 15 (10.3) 2 (1.4)	56 (20.7) 38 (14.0) 16 (5.9) 2 (0.7)
**Most common treatment-related TEAEs by SOC (reported in ≥5% of any subgroup or overall), and classified by HLGT ^b^ or PT ^c^** **Blood and lymphatic system disorders** Anemia ^c^ Anemia of malignant disease ^c^ Febrile neutropenia ^c^ Neutropenia ^c^ Thrombocytopenia ^c^ **Vascular disorders** Hypertension ^c^ Hypertensive crisis ^c^ Hypotension ^c^ Thrombosis ^b^ **Respiratory, thoracic and mediastinal disorders** Acute respiratory failure ^c^ Bronchial disorders ^b^ Dyspnea ^c^ Lower respiratory tract disorders (excluding obstructions and infections) ^b^ Lung disorder ^c^ Pulmonary embolism ^c^ **Cardiac disorders** Cardiac arrhythmia ^b^ Coronary artery disorders ^b^ Heart failures ^b^ **Infections and infestations** Ancillary infectious topics ^b^ Bacterial infectious disorders ^b^ Infections, unspecified ^b^ Viral infectious disorders ^b^	**2 (3.6)** 2 (3.6) 1 (1.8) 0 (0.0) 0 (0.0) 2 (3.6) **2 (3.6)** 1 (1.8) 0 (0.0) 0 (0.0) 2 (3.6) **2 (3.6)** 0 (0.0) 2 (3.6) 1 (1.8) 0 (0.0) 0 (0.0) 0 (0.0) **0 (0.0)** 0 (0.0) 0 (0.0) 0 (0.0) **0 (0.0)** 0 (0.0) 0 (0.0) 0 (0.0) 0 (0.0)	**4 (5.7)** 2 (2.9) 0 (0.0) 0 (0.0) 1 (1.4) 4 (5.7) **4 (5.7)** 4 (5.7) 0 (0.0) 0 (0.0) 0 (0.0) **3 (4.3)** 0 (0.0) 0 (0.0) 2 (2.9) 1 (1.4) 0 (0.0) 0 (0.0) **0 (0.0)** 0 (0.0) 0 (0.0) 0 (0.0) **4 (5.7)** 1 (1.4) 1 (1.4) 1 (1.4) 1 (1.4)	**13 (8.9)** 5 (3.4) 0 (0.0) 1 (0.7) 3 (2.1) 8 (5.5) **7 (4.8)** 6 (4.1) 1 (0.7) 1 (0.7) 0 (0.0) **8 (5.5)** 1 (0.7) 0 (0.0) 2 (1.4) 3 (2.1) 1 (0.7) 1 (0.7) **11 (7.5)** 4 (2.7) 1 (0.7) 6 (4.1) **6 (4.1)** 0 (0.0) 2 (1.4) 4 (2.7) 0 (0.0)	**19 (7.0)** 9 (3.3) 1 (0.4) 1 (0.4) 4 (1.5) 14 (5.2) **13 (4.8)** 11 (4.1) 1 (0.4) 1 (0.4) 2 (0.7) **13 (4.8)** 1 (0.4) 2 (0.7) 5 (1.8) 4 (1.5) 1 (0.4) 1 (0.4) **11 (4.1)** 4 (1.5) 1 (0.4) 6 (2.2) **10 (3.7)** 1 (0.4) 3 (1.1) 5 (1.8) 1 (0.4)

Data presented as n (%). n represents the number of patients who experienced one or more AEs. Patients were counted only once for each PT, HLGT or SOC. The total number at the SOC level may be lower than the sum of the individual numbers reported at HLGT or PT level, because one patient could experience multiple events. AEs were coded using MedDRA version 23.0 and graded using NCI-CTCAE version 4.03. ^a^ Fatal treatment-related TEAEs were due to cardiac disorders (one fatal cardiac arrhythmia; one fatal coronary artery disorder). ^b^ Treatment-related TEAE HLGT classification. ^c^ Treatment-related TEAE PT classification. 2L = second line; 3L = third line; 4L+ = fourth or later lines; AE = adverse event; CTCAE = Common Terminology Criteria for Adverse Events; HLGT = High-Level Group Term; PT = Preferred-Term; SAE = serious AE; SOC = System Organ Class; TEAE = treatment-emergent AE.

## Data Availability

The data that support the findings of this study are available from the corresponding author upon reasonable request. Qualified researchers may request data from Amgen studies. Complete details are available at the following: http://www.amgen.com/datasharing.

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
