# Peer review of "Real-World Use and Effectiveness of Carfilzomib Plus Dexamethasone in Relapsed/Refractory Multiple Myeloma in Europe"

_cancers, 2022, doi:10.3390/cancers14215311_

Round 1

Reviewer 1 Report

Congratulations on this work. Minor comments are suggested.

Author Response

"Please see attachment"

Reviewer 2 Report

This paper is well written and demonstrates the usefulness of Kd therapy for RRMM in real-world clinical practice through a multicenter study. In particular, the authors found that Kd therapy showed acceptable therapeutic response in RRMM, regardless of line of therapy.

1. The reason why Kd was chosen over KRd in this cohort could be stated a little more clearly. In Table 3, do the authors have information on how many patients were included whose Kd was chosen because of cytopenias or renal dysfunction?

2. Cardiac disease appears to be infrequent in Table 5, except for 4L+RPMM patients. Should this mean that cardiac events due to Kd do not need to be closely watched in clinical practice, or should it be considered that Kd is difficult to use in patients with significant cardiac disease?

3. I would like to suggest that the authors show what profile RRMM patients should receive Kd therapy.
